# Graph Structural Aggregation for Explainable Learning

## Abstract

Graph neural networks have proven to be very efficient to solve several tasks in graphs such as node classification or link prediction. These algorithms that operate by propagating information from vertices to their neighbors allow one to build node embeddings that contain local information. In order to use graph neural networks for graph classification, node embeddings must be aggregated to obtain a graph representation able to discriminate among different graphs (of possibly various sizes). Moreover, in analogy to neural networks for image classification, there is a need for explainability regarding the features that are selected in the graph classification process. To this end, we introduce StructAgg, a simple yet effective aggregation process based on the identification of structural roles for nodes in graphs that we use to create an end-to-end model. Through extensive experiments we show that this architecture can compete with state-of-the-art methods. We show how this aggregation step allows us to cluster together nodes that have comparable structural roles and how these roles provide explainability to this neural network model.

## 1 Introduction

Convolution neural networks (LeCun et al., 1995) have proven to be very efficient at learning meaningful patterns for many articicial intelligence tasks. They convey the ability to learn hierarchical information in data with Euclidean grid-like structures such as images and text. Convolutional Neural Networks (CNNs) have rapidly become state-of-the art methods in the fields of computer vision (Russakovsky et al., 2015) and natural language processing (Devlin et al., 2018).

However in many scientific fields, studied data have an underlying graph or manifold structure such as communication networks (whether social or technical) or knowledge graphs. Recently there have been many attempts to extend convolutions to those non-Euclidean structured data (Hammond et al., 2011; Kipf & Welling, 2016; Defferrard et al., 2016). In these new approaches, the authors propose to compute node embeddings in a semi-supervised fashion in order to perform node classification. Those node embeddings can also be used for link prediction by computing distances between each node of the graph (Hammond et al., 2011; Kipf & Welling, 2016).

Graph classification is studied in many fields. Whether for predicting the chemical activity of a molecule or to cluster authors from different scientific domains based on their ego-networks (Freeman, 1982). However when trying to generalize neural network approaches to the task of graph classification there are several aspects that differ widely from image classification. When trying to perform graph classification, we can deal with graphs of different sizes. To compare them we first need to obtain a graph representation that is independant of the size of the graph. Moreover, for a fixed graph, nodes are not ordered. The graph representation obtained with neural network algorithms must be independant of the order of nodes and thus be invariant by node permutation.

Aggregation functions are functions that operate on node embeddings to produce a graph representation. When tackling a graph classification task, the aggregation function used is usually just a mean or a max of node embeddings as illustrated in figure 1b. But when working with graphs of large sizes, the mean over all nodes does not allow us to extract significant patterns with a good discriminating power. In order to identify patterns in graphs, some methods try to identify structural roles for nodes. Donnat et al. (2018) define structural role discovery as the process of identifying nodes which have topologically similar network neighborhoods while residing in potentially distant areas of the network

as illustrated in figure 1a. Those structural roles represent local patterns in graphs. Identifying them and comparing them among graphs could improve the discriminative power of graph embeddings obtained with graph neural networks. In this work, we build an aggregation process based on the identification of structural roles, called StructAgg.

The main contributions of this work are summarized bellow:

1. **Learned aggregation process.** A differentiable aggregation process that learns how to aggregate node embeddings in order to produce a graph representation for a graph classification task.

2. **Identification of structural roles.** Based on the definition of structural roles from Donnat et al. (2018), our algorithm learns structural roles during the aggregation process. This is innovative because most algorithms that learn structural roles in graphs are not based on graph neural networks.

3. **Explainability of selected features for a graph classification task.** The identification of structural roles enables us to understand and explain what features are selected during training. Graph neural networks often lack explainability and there are only few works that tackle this issue. One contribution of this work is the explainability of the approach. We show how our end-to-end model provides interpretability to a graph classification task based on graph neural networks.

4. **Experimental results.** Our method achieves state-of-the-art results on benchmark datasets. We compare it with kernel methods and state-of-the-art message passing algorithms that use pooling layers as aggregation processes.

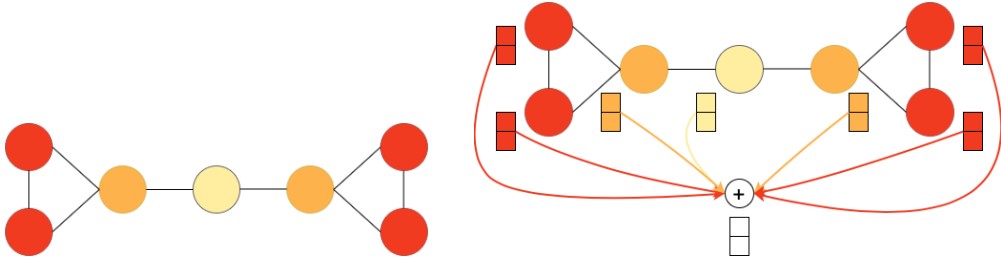

(a) nodes with the same structural role are classified together (same color).

(b) aggregation process to create a graph embedding, the node features are summed to produce a representation of the graph.

Figure 1: Identification of structural roles and aggregation of node features over the whole graph.

## 2 RELATED WORK

The identification of nodes that have similar structural roles is usually done by an explicit featurization of nodes or by algorithms that rely on random walks to explore nodes' neighborhoods. A well known algorithm in this line of research is RolX (Gilpin et al., 2013; Henderson et al., 2012), a matrix factorization that focuses on computing a soft assignment matrix based on a listing of topological properties set as inputs for nodes. Similarly struct2vec builds a multilayered graph based on topological metrics on nodes and then generates random walks to capture structural information. In another line of research, many works rely on graphlets to capture nodes' topological properties and identify nodes with similar neighborhoods (Rossi et al., 2017; Lee et al., 2018; Ahmed et al., 2018). In their work, Donnat et al. (2018) compute node embeddings from wavelets in graphs to caracterize nodes' neighborhood at different scales.

In this work, we introduce an aggregation process based on the identification of structural roles in graphs that is computed in an end-to-end trainable fashion. We build a hierarchical representation of nodes by using neural network models in graphs to propagate nodes' features at different hops. Recently there has been a rich line of research, inspired by deep models in images, that aims at

redefining neural networks in graphs and in particular convolutional neural networks (Defferrard et al., 2016; Kipf & Welling, 2016; Veličković et al., 2017; Hamilton et al., 2017; Bronstein et al., 2017; Bruna et al., 2013; Scarselli et al., 2009). Those convolutions can be viewed as message passing algorithms that are composed of two phases. A message passing phase that runs for T steps is first defined in terms of message functions and vertex update functions. A readout phase then computes a feature vector for the whole graph using some readout function. In this work we will see how to define a readout phase that is learnable and that is representative of meaningful patterns in graphs.

## 3    PROPOSED METHOD

In this section we introduce the structural aggregation layer (StructAgg). We show how we identify structural classes for nodes in graphs; how those classes are used in order to develop an aggregation layer; and how this layer allows us to compare significant structural patterns in graphs for a supervised classification task.

### 3.1    NOTATIONS

Let $G = (V, E, X)$ be a graph, where $V$ is the set of nodes of $G$, $E$ the set of edges and $X \in \mathbb{R}^{n \times f}$ the feature matrix of $G$'s nodes where $f$ is the dimensionality of node features. Let $n = |V|$ be the number of nodes of $G$ and $e = |E|$ the number of edges of $G$.

Let $A$ be the adjacency matrix of graph $G$ and $D$ be its degree diagonal matrix. Let $v_i$ and $v_j$ be the $i^{th}$ and $j^{th}$ nodes of $G$, we have:

$$A_{ij} = \left\{ \begin{array}{ll} 1 & \text{if } (v_i, v_j) \in \text{E} \\ 0 & \text{otherwise} \end{array} \right. , D_{ii} = \sum_j A_{ij}$$

Let $S = \{G_1, ..., G_d\}$ be a set of $d$ graphs and $\{y_1, ..., y_d\}$ be the labels associated with these graphs.

### 3.2    HIERARCHICAL STRUCTURAL EMBEDDING

**Graph neural networks.** We build our work upon graph neural networks (GNNs). Several architectures of graph neural networks have been proposed by Defferrard et al. (2016); Kipf & Welling (2016); Veličković et al. (2017) or Bruna & Li (2017). Those graph neural network models are all based on propagation mechanisms of node features that follow a general neural message passing architecture (Ying et al., 2018; Gilmer et al., 2017):

$$X^{(l+1)} = MP(A, X^{(l)}; W^{(l)}) \tag{1}$$

where $X^{(l)} \in \mathbb{R}^{n \times f_l}$ are the node embeddings computed after $l$ steps of the GNN, $X^{(0)} = X$, and $MP$ is the message propagation function, which depends on the adjacency matrix. $W^{(l)}$ is a trainable weight matrix that depends on layer $l$. Let $f_l$ be the dimension of the node vectors after $l$ steps of the GNN, $f_0 = f$.

The aggregation process that we introduce next can be used with any neural message passing algorithm that follows the propagation rule 1. In all the following of our work we denote by $MP$ the algorithm. For the experiments, we consider Graph Convolutional Network (GCN) defined by (Kipf & Welling, 2016). This model is based on an approximation of convolutions on graphs defined by (Defferrard et al., 2016) and that use spectral decompositions of the Laplacian. The popularity of this model comes from its computational efficiency and the state-of-the-art results obtained on benchmark datasets. This layer propagates node features to 1-hop neighbors. Its propagation rule is the following:

$$X^{(l+1)} = MP(A, X^{(l)}; W^{(l)}) = GCN(A, X^{(l)}) = \rho(\tilde{D}^{-1/2} \tilde{A} \tilde{D}^{-1/2} X^{(l)} W^{(l)}) \tag{2}$$

Where $\rho$ is a non-linear function (a $ReLU$ in our case), $\tilde{A} = A + I_n$ is the adjacency matrix with added self-loops and $\tilde{D}_{ii} = \sum_j \tilde{A}_{ij}$.

This propagation process allows us to obtain a node representation representing its $l$-hop neighborhood after $l$ layers of GCN. We build a hierarchical representation for nodes by concatenating their embeddings after each step of GCN. The final representation $X_{structi}$ of a node $i$ is given by:

$$X_{structi} = \coprod_{l=1}^{L} X_i^{(l)} \tag{3}$$

Where $L$ is the total number of GCN layers applied.

**Identifying structural classes.** Embedding nodes with $MP$ creates embeddings that are close for nodes that are structurally equivalent. Some use a handcrafted node embedding based on propagation processes with wavelets in graphs to identify structural clusters based on hierarchical representation of nodes (Donnat et al., 2018). By analogy, we learn hierarchical node embeddings and an aggregation layer that identifies structural roles for node in graphs. Those structural roles are consistent along graphs of a dataset which allows us to bring interpretability to our graph classification task.

Node features $X_{struct}$ contain the information of their $L$-hop neighborhood decomposed into $L$ vectors each representing their $l$-hop neighborhood for $l$ varying between 1 and $L$. We will show next that nodes that have the same $L$-hop neighborhood are embedded into the same vector.
To identify structural roles, we thus project each node embedding on a matrix $p \in \mathbb{R}^{f_{struct} \times c}$ where $c$ is the number of structural classes and $f_{struct} = \sum_{l=1}^{L} f_l$ is the dimensionality of $X_{structi}$ for each node $i$. We obtain a soft assignment matrix:

$$C = softmax(X_{struct}p) \in \mathbb{R}^{n \times c} \tag{4}$$

Where the softmax function is applied in a row-wise fashion. This way, $C_{ij}$ represents the probability that node $i$ belongs to cluster $j$.

**Definition 1.** *Let $i$ and $j$ be two nodes of a graph $G = (V, E, X)$. Let $\mathcal{N}_l(i) = \{i' \in N | d(i, i') \leq l\}$ be the $l$-hop neighborhood of $i$, which means all the nodes that are at distance lower of equal to $l$ of $i$, $d$ being the shortest-path distance. Let $X_{\mathcal{N}_l(i)}$ be the feature matrix of the $l$-hop neighborhood of $u$. Let $G_{i,l}$ be the subgraph of $G$ composed of the $l$-hop neighborhood of $i$.*
*We say that $i$ and $j$ are $l$-structurally equivalent if there exists an isomorphism $\psi$ from $\mathcal{N}_l(j)$ to $\mathcal{N}_l(i)$ such that the two following conditions are verified:*

- $G_{i,l} = \psi(G_{j,l})$

- $\forall j' \in \mathcal{N}_l(j), X_{\psi(j')} = X_{j'}$

**Theorem 1.** *Two nodes $i$ and $j$ that are $L$-structurally equivalent have the same final embedding, $X_{structi} = X_{structj}$.*

### 3.3 STRUCTURAL AGGREGATION

After having identified structural classes, we aggregate node embeddings over those classes, as illustrated in figure 2. The goal is to obtain an embedding that discriminates graphs that do not belong to the same class and that selects similar patterns in graphs of the same class. Graphs that have similar properties and thus similar node patterns should have nodes with similar roles and similar embeddings.

**Performing a structural aggregation.** The structural aggregation aims at comparing embeddings of nodes that have similar roles in the graph. When computing the distance between two graphs, if those two graphs have the same distribution of structural roles, they will have embeddings that are close. Nodes that are leaves or nodes that are central in the graph should be compared separately.

Mathematically, the variance over all nodes if the graph may be high and decomposing nodes per structural roles aims at decreasing the variance per cluster and thus at bringing more information to the final graph embedding.

The structural aggregation layer performs an aggregation per structural role. This aggregation is a mean of embeddings of nodes that belong to the same cluster. The final embedding is a concatenation of the mean embeddings of the nodes per cluster of structural role.

$$Z_{graph} = C^T X^{(L)} \in \mathbb{R}^{c \times f_L}$$

**Proposition 1.** *The embedding $Z_{graph}$ is invariant by node permutation.*

Moreover when performing soft classification, the output cluster assignment for each node should generally be close to a one-hot vector so that each node has a clear structural role identified. We therefore regularize the entropy of the cluster assigmnent matrix by minimizing $L_{reg} = \frac{1}{n} \sum_{i=1}^{n} H(C_{i.})$ where $H(C_{i.})$ is the entropy of the assignment vector $C_{i.}$ of node $i$. This is often done when identifying classes for nodes. The same regularization is applied in (Ying et al., 2018) to identify communities in graphs to develop a pooling layer.

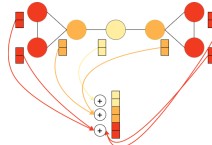

Figure 2: Aggregation of node features over structural classes

**Remarks.**

- The structural classes identified by our algorithm are local. They contain the information of the $L$-hop neighborhood of each node.

- Compared to the structural embedding presented in Donnat et al. (2018), our algorithm needs multiple graphs and is trained in a supervised fashion. Donnat et al. (2018) introduced a node embedding that allows us to identify structural classes in a single graph. It is not straightforward how to generalize this procedure to the case of a set of graphs. Indeed, when dealing with a single graph, this procedure is very efficient at identifying structural roles. However, those structural roles are defined per graph and thus two nodes that have the same structural role but that lie in two different graphs can have embeddings that differ and can thus be classified in two different classes.

### 3.4 INTERPRETABLE GRAPH NEURAL NETWORK

Graph neural networks lack interpretability. In the case of convolutional neural networks on images, it is possible to visualize the activation of certain layers to have an idea of the patterns that are selected during the process of image classification and of which pattern is usefull to discriminate images of a certain class from the rest of the images of a dataset. In the case of graphs, most algorithms based on neural network models bring no interpretability regarding the features that are selected and that make the classification accurate.

In this work, we bring some interpretability to this class of models and illustrate it in the next sections in experiments. By propagating information from node to node, we are able to identify structural roles in graphs. Those structural roles contain the information of local neighborhoods of nodes and of their local topological structure. By identifying roles in graphs, we compare between graphs the embeddings per structural roles. This way, we can identify roles that are specific to a certain class and caracterize each class with the combination of roles they are made of. We show through extensive experiments that the information contained in the identification of roles allows us to discriminate graphs of different classes and lead the way to other works that could bring interpretability to this field of research.

## 4 EXPERIMENTS

### 4.1 GRAPH CLASSIFICATION

**Datasets:** We choose a wide variety of benchmark datasets for graph classification to evaluate our model. The datasets can be separated in two types. 2 bioinformatics datasets: PROTEINS and

D&D; and a social network dataset: COLLAB. In the bioinformatics datasets, graphs represent chemical compounds. Nodes are atoms and edges represent connections between two atoms. D&D and PROTEINS contain two classes of molecules that represent the fact that a molecule can be either active or inactive against a certain type of cancer. The aim is to classify molecules according to their anti-cancer activity. COLLAB is composed of ego-networks. Graphs' labels are the nature of the entity from which we have generated the ego-network. In Table 1 we report some information on these datasets such as the maximum number of nodes in graphs, the average number of nodes per graph, the number of graphs, the size of node features $f$ (if available) and the number of classes. More details can be found in (Yanardag & Vishwanathan, 2015). We also use the new database Open Graph Benchmark (OBG) to test our method on larger datasets (Hu et al., 2020). We use the ogb-molhiv dataset whose properties are listed bellow. Each graph represents a molecule, where nodes are atoms, and edges are chemical bonds. Input node features are 9-dimensional, containing atomic number and chirality, as well as other additional atom features such as formal charge and whether the atom is in the ring or not.

**Experimental setup:** We perform a 10-fold cross validation split which gives 10 sets of train, validation and test data indices in the ratio 8:1:1. We use stratified sampling to ensure that the class distribution remains the same across splits. We fine tune hyperparameters $n_{roles}$ the number of structural roles, $lr$ the learning rate and finally the dimensions of the successive layers respectively chosen from the sets $\{2, 5, 10, 20\}$, $\{0.01, 0.001, 0.0001\}$, $\{8, 16, 32, 64, 128, 256\}$. The sets from which our hyperparameters are selected vary according to the sizes of graphs in each dataset. We do not set a maximum number of epochs but we perform early stopping to stop the training which means that we stop the training when the validation loss has not improved for 20 epochs. We report the mean accuracy and the standard deviation over the 10 folds on the validation set. We compare our method with kernel methods and with a graph neural network that uses pooling layers (Ying et al., 2018). We should note that kernel methods do not use node features that are available on bioinformatics datasets. For COLLAB, we don't have any features available on nodes. We compute the one-hot encoding of node degrees that we use as node features for our algorithm.

**Results:** From the results of Table 1 we can observe that StructAgg is competing with state-of-the-art methods. Indeed, on most datasets, StructAgg has a score very close to those obtained by Ying et al. (2018) and Gao & Ji (2019). From Table 1, StructAgg outperforms all algorithms on COLLAB. Moreover, StructAgg allows us to improve classification results from GCN on ogb-molhiv dataset as illustrated in figure 2

| Dataset | D&D | PROTEINS | COLLAB |
|---|---|---|---|
| Max #Nodes | 5748 | 620 | 492 |
| Avg #Nodes | 284.32 | 39.06 | 74.49 |
| #Graphs | 1178 | 1113 | 5000 |
| $f$ | 89 | 3 | - |
| classes | 2 | 2 | 3 |
| Graphlet (Shervashidze et al., 2009) | 74.85 | 72.91 | 64.66 |
| Shortest-Path (Borgwardt & Kriegel, 2005) | 78.86 | 76.43 | 59.10 |
| 1-WL (Shervashidze et al., 2011) | 74.02 | 73.76 | 78.61 |
| WL-OA (Kriege et al., 2016) | 79.04 | 75.26 | 80.74 |
| GraphSage (Hamilton et al., 2017) | 75.42 | 70.48 | 68.25 |
| DGCNN (Zhang et al., 2018) | 79.37 | 76.26 | 73.76 |
| DIFFPOOL (Ying et al., 2018) | 80.64 | 76.25 | 75.48 |
| g-U-Nets (Gao & Ji, 2019) | **82.43** | **77.68** | 77.56 |
| **StructAgg** | $78.42 \pm 0.97$ | $76.72 \pm 2.53$ | $\mathbf{80.26 \pm 2.73}$ |

Table 1: Classification accuracy on bioinformatics datasets

| Name | #Graphs | #Node per Graph | #Edges per Graph | GCN | strcutAgg |
|------|---------|-----------------|------------------|-----|-----------|
| ogbg-molhiv | 41127 | 25.5 | 27.5 | $0.7606 \pm 0.0097$ | $0.7701 \pm 0.0102$ |

Table 2: OGB dataset for graph classification. The score reported is the ROC-AUC score.

## 4.2 IDENTIFICATION OF STRUCTURAL ROLES

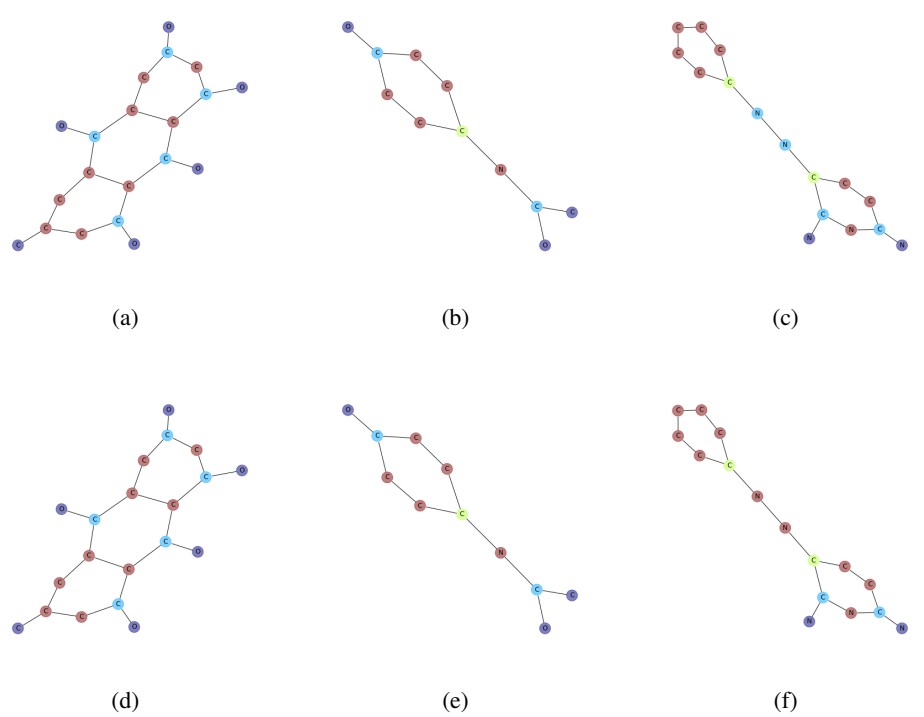

Figure 3: Molecules drawn for a bioinformatics dataset. The colors represent structural classes identified by our algorithm in figures 3a, 3b and 3c compared to the output of GraphWave in figures 3d, 3e and 3f. To compare the structural classes output by the two algorithms, 3a and 3d are the same molecule, 3b and 3e also and 3c and 3f are the same molecule. To obtain comparable results, we run our algorithm and output the structural classes for each molecule. For each molecule, we then run GraphWave with the number of classes from StructAgg being set as input of GraphWave.

We show some examples of molecules of a molecular dataset and the roles identified by our algorithms. From figure 3 we can see that there is some consistence in the assignment of nodes to structural roles. Moreover, a great advantage of our method is that it takes into account node features that are in our case a one-hot encoding of the atom type. Compared to Donnat et al. (2018) whose embeddings do not include node features, our method identifies roles in a macroscopic fashion by identifying similar roles in the whole dataset and not in a single graph. Most methods rely on computed features that correspond to a single graph and do not generalize to multiple graphs because of scales issues. It is the case for GraphWave for which features computed in order to cluster nodes per roles depend on the size of the graph. To compare our role assignment with GraphWave, we computed assignments per graph and not along the whole dataset. We can note that compared to GraphWave, our algorithm uses node features to select structal roles.

### 4.3 Importance of structural patterns

Identifying roles in graphs boils down to identifying significant patterns. As shown in subsection 4.2, the roles that we were able to identify represent coherent patterns in graphs. Graphs' final embeddings are made of two parts. Nodes' embeddings that are the result of successive GCNs and the allocation of all nodes to different structural classes. We would like to quantify how much information is contained in the structural roles and if the decomposition of the graph in $c$ classes improves our classification accuracy. To this end, from our trained algorithm, we compute all allocation matrices of all graphs of a dataset.

In order to validate the fact that roles themselves have a high discriminative power, we need to identify combinations of roles that are specific to different classes. We thus want to identify the structural roles that compose each graph.

Let's consider a graph $G$ and an allocation matrix $C \in \mathbb{R}^{n \times c}$. We obtain a soft assignment matrix. We compute on it a histogram per class over each node of the graph. The feature vector is now a concatenation of all histograms per class. We have:

$$Feat_G = \overset{c}{\underset{j=1}{\Big\|}} \ hist(C_{.j}, bins) \in \mathbb{R}^{bins*c}$$

Where $bins$ is the number of bins for the histogram. The histogram upper and lower bounds are $0$ and $1$ because the values of $C$ are the output of a softmax function.

Let $(G_1, ..., G_d)$ be $d$ graphs of labels $(y_1, ..., y_d)$. After having computed all graphs' histograms we obtain $d$ vectors $(Feat_{G_1}, ..., Feat_{G_d})$ that we use as imputs of a classifier. We use a SVM and we display the results in table 3 in order to compare the information that comes from the strutural roles identification and the information that comes from node embeddings.

We can see that the accuracy is lower when we don't consider node embeddings which is consistent with the fact that a lot of information is contained in the embeddings of nodes that are the outputs of GCNs. But the accuracy is significantly higher than a random model which proves that we can identify some patterns in the distribution of structural roles among graphs and that those patterns are a good first approximation to separate classes in a dataset.

| Dataset | D&D | PROTEINS | COLLAB |
|---|---|---|---|
| **StructAgg** | $78.42 \pm 0.97$ | $76.72 \pm 2.53$ | $80.26 \pm 2.73$ |
| **StructHist** | $74.54 \pm 2.87$ | $73.68 \pm 2.03$ | $70.94 \pm 2.08$ |

Table 3: Classification accuracy based on the histogram of the assignment matrix (StructHist) compared to our algorithm (StructAgg).

## 5 Conclusion

Graph neural networks are very effective to build node embeddings by propagating node features in graphs. But in order to build a graph representation from these embeddings there are several ways to procede. In this work, we proposed a novel technique based on the identification of structural roles for nodes in graphs. We showed that the identification of roles allows us to compare patterns between graphs that are significant for graph classification. Moreover, this work opens new perspectives in the field of explainability of graph neural networks. We showed that these patterns brought explainability to this task and to which kind of structures are selected by our algorithm during the process of graph classification. A better understanding of which features are selected during training can enable us to develop new methods based on drawn conclusion and maybe open new perspective in applied mathematics for drug discovery.

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

## A  APPENDIX

You may include other additional sections here.

## B  PROOFS

### B.1  PROOF OF THEOREM 1

**Theorem 2.** *Two nodes $i$ and $j$ that are $L$-structurally equivalent have the same final embedding, $X_{struct\,i} = X_{struct\,j}$.*

*Proof.* Let $i$ and $j$ be two nodes of a graph $G = (V, E, X)$ that are $L$-structurally equivalent.
Let $\mathcal{P}(l)$ be the following proposition:
Two nodes that are $l$-structurally equivalent for some $l$, have the same embedding after $l$ steps of GCN.
Let's prove this proposition by induction.
$\mathcal{P}(0)$ is true, two nodes that have the same embedding are $0$-structurally equivalent after $0$ step of GCN.
Let $\mathcal{P}(l)$ be true and let's prove $\mathcal{P}(l+1)$.
Let $i$ and $j$ be two nodes that are $l$-structurally equivalent. After a step of GCN we have:

$$X^{(l+1)} = GCN(A, X^{(l)})$$

So we have:

$$(\tilde{D}^{-1/2}\tilde{A}\tilde{D}^{-1/2}X^{(l)})_i = \sum_{i' \in \mathcal{N}(i)} \frac{a_{ii'}}{\sqrt{d_i d_{i'}}} X^{(l)}_{i'}$$

Since $i$ and $j$ are $l$-structurally equivalent, there exists an isomorphism $\psi$ such that:

$$\forall i' \in \mathcal{N}_l(i), \exists j' \in \mathcal{N}_l(j) \text{ such that } X^{(l)}_{i'} = X^{(l)}_{\psi(j')} = X^{(l)}_{j'}$$

$$\Rightarrow \sum_{i' \in \mathcal{N}(i)} \frac{a_{ii'}}{\sqrt{d_i d_{i'}}} X^{(l)}_{i'} = \sum_{j' \in \mathcal{N}(j)} \frac{a_{\psi(j)\psi(j')}}{\sqrt{d_{\psi(j)} d_{\psi(j')}}} X^{(l)}_{\psi(j')} = \sum_{j' \in \mathcal{N}(j)} \frac{a_{jj'}}{\sqrt{d_j d_{j'}}} X^{(l)}_{j'}$$

$$\Rightarrow (\tilde{D}^{-1/2}\tilde{A}\tilde{D}^{-1/2}X^{(l)})_i = (\tilde{D}^{-1/2}\tilde{A}\tilde{D}^{-1/2}X^{(l)})_j$$

$$\Rightarrow X^{(l+1)}_i = f(\tilde{D}^{-1/2}\tilde{A}\tilde{D}^{-1/2}X^{(l)}W^{(l)})_i$$

$$= f(\tilde{D}^{-1/2}\tilde{A}\tilde{D}^{-1/2}X^{(l)}W^{(l)})_j$$

$$= X^{(l+1)}_j$$

So two nodes $i$ and $j$ that are $L$ structurally equivalent, are $l$ structurally equivalent for all $l$ between $0$ and $L$ and thus, $X_{struct\,i} = X_{struct\,j}$ because $X_{struct}$ is the concatenation of embeddings after each layer. $\qquad \square$

## B.2 PROOF OF PROPOSITION 2

**Proposition 2.** *The embedding $Z_{graph}$ is invariant by node permutation.*

*Proof.* Let $P \in {0,1}^{n \times n}$ be any permutation matrix. Since $P$ is a permutation matrix we have $PP^T = I$. We have $PX^{(l+1)} = GCN(PAP^T, PX^{(l)})$.

$$Z_{graph} = C^T X^{(L)}$$
$$= (softmax(X_{struct}p))^T X^{(L)}$$
$$= (softmax(P^T P X_{struct}p))^T P^T P X^{(L)}$$

Since the softmax is applied in a row-wise fashion, we have:

$$Z_{graph} = (P^T softmax(P X_{struct}p))^T P^T P X^{(L)}$$
$$= (softmax(P X_{struct}p))^T P X^{(L)}$$
$$= Z_{P graph}$$

Where $Z_{P graph}$ is the embedding of our graph to which we have applied the permutation $P$ $\qquad \square$

