# OpenReview forum: "Graph Structural Aggregation for Explainable Learning"
_ICLR.cc/2021/Conference — Reject_

### Official Review · AnonReviewer4 · 2020-10-26
**StructAgg: an ultra-high dimensional node representation in graph neural network that preserves local graph structure for better explanation**

**Rating:** 6
**Confidence:** 5

**Review:**

This paper proposed the StructAgg, an aggregation algorithm in convolutional graph neural network that learns the structural roles for nodes in the graph embedding. In this algorithm, a structural representation of node is constructed through concatenation of latest p layers of node presentation in graph neural network, which consists of information from the p-hop neighborhood. The paper is good quality and its demonstration is clear.  The idea behind is original. However, learning embeddings through concatenation of multiple layers of neural representation is not completely new. The contribution of this paper to the community is not ground-breaking as well.  From the experiment results, it is hard to stress its significance as in most of times this method does not beat the state-of-the-art algorithms.

Pros:
1.  As opposed to previous studies, this paper focus on learning node embeddings that preserve local information in a larger p-hop neighborhood. Such idea assumes that the node structural role can be identified via its neighborhood and the node embeddings should carry information regarding its structural roles. To achieve that, the learning algorithms should balance the predictability of the embedding in the node classification as well as the consistency of node representation towards underlying clustering in the graph.
2.  The proposed representation learned by StructAgg, is able to provide the clustering of nodes as well as the node embeddings that are invariant to the permutation.
3.  The StructAgg only need to concatenate the existing learned representation in lower layers, thus it is easy to implement.

Cons:
1.  Ulta-high dimensional representation as the size of local neighborhood of interest increases: The concatenation operation expands the dimensionality of embeddings drastically as the local neigborhood expands. As a result, the StructAgg representation can only feasibly represent a structures that are present in small neighborhood graph. In some graph like trees, it is fine. But in complex graphs, with larger local structures, it is not efficient in both storage and computation.
2. Tradeoff between the classification and clustering:  The proposed method defines the loss function as the combination of supervised node classification loss and unsupervised node cluster assignment loss (in terms of entropy of softmax assignment). The formation of this joint loss function implies that the node label is assigned consistently according to the structural role of the node in the graph with the number of total structural classes in the graph known. It can be imagined that with the number of structural classes increases, the label assignment may not be consistent and will result in decrease of performance. It seems that this method is more suitable for two or three structural classes as shown in the experiment.
3.  Lack of robustness: The message passing algorithm in small neighborhood tends to preserve not just local information and local noises. In StructAgg, since the aggregation happens only within the same cluster, the noise level in different clusters may varies a lot, depending on the size of cluster. It is more likely to create noisy embeddings in this small clusters and present them in the final embeddings, while in traditional aggregation, such noise level is suppressed due to the involvement of more nodes.

Some question need to clarify:
1.  In proof of theorem 1, please explain for $\sum_{i^{'}\in \mathcal{N}(i)}\frac{a_{i,i^{'}}}{\sqrt{d_{i}d_{i^{'}}}}X_{i^{'}}^{(l)} = \sum_{j^{'}\in \mathcal{N}(j)}\frac{a_{j,j^{'}}}{\sqrt{d_{j}d_{j^{'}}}}X_{\psi(j^{'})}^{(l)}$, what is the relation between coefficients $\frac{a_{i,i^{'}}}{\sqrt{d_{i}d_{i^{'}}}}$ and $\frac{a_{j,j^{'}}}{\sqrt{d_{j}d_{j^{'}}}}$. Also should it be like $\sum_{j^{'}\in \mathcal{N}(j)}\frac{a_{\psi(j),\psi(j^{'})}}{\sqrt{d_{\psi(j)}d_{\psi(j^{'})}}}X_{\psi(j^{'})}^{(l)}$ ?

2. In proof of proposition 2, please explain in detail the reason behind $(softmax(P^{T}PX_{struct}p))^{T}P^{T}PX^{L} = (P^{T}softmax(PX_{struct}p))^{T}PP^{T}PX^{L}$. Here i believe it should be $(softmax(P^{T}PX_{struct}p))^{T}P^{T}PX^{L} = (P^{T}softmax(PX_{struct}p))^{T}P^{T}PX^{L}$, with a mistakenly added $P$ in original draft.  The reason is that the softmax is row operation, a row permutation $P^{T}$ does not affect the result of $softmax(\cdot)$ at each row but their ordering. So  $softmax(P^{T}\cdot)= P^{T}softmax(\cdot)$, thus $(softmax(P^{T}PX_{struct}p)) = (P^{T}softmax(PX_{struct}p))$. This leaves
$(softmax(P^{T}PX_{struct}p))^{T}P^{T}PX^{L} = (P^{T}softmax(PX_{struct}p))^{T}P^{T}PX^{L}
= (softmax(PX_{struct}p))^{T}PP^{T}PX^{L}  = (softmax(PX_{struct}p))^{T}PX^{L} $

---

### Official Review · AnonReviewer2 · 2020-10-27
**Review comments to Paper 641**

**Rating:** 4
**Confidence:** 4

**Review:**

==========Summary==========

In this paper, the authors investigate how to improve pooling functions in graph neural networks for the purpose of better addressing graph classification problems. The core idea in this paper is to group node representations. In particular, the authors develop StructAgg, a softmax based implementation, to parameterize the grouping process, and the parameters in StructAgg can be learned from downstream supervision signals. Empirical results focus on three aspects: (1) Improvement in graph classification accuracy brought by StructAgg; (2) Visualization on grouped node representations; and (3) Comparison between two variants of StructAgg.

==========Reason for the rating==========

For the current draft, I am leaning to reject. Although pooling for graph classification is a meaningful problem, it is difficult to see the unique perspective or value in the proposed technique compared with existing approaches. For now, the technical depth in this paper seems to be limited, and more convincing empirical evidences are expected.

==========Strong points==========
1. This paper delves into a meaningful problem. Indeed, pooling functions in graph neural networks are critical for graph classification tasks. While existing solutions are simple and efficient, they may be sub-optimal in some cases.

2. The authors propose a grouping-based method to reduce variance in graph representations, which potentially could improve generalization performance.

3. Empirical evidences are provided to confirm the effectiveness and impact from StructAgg.

==========Weak points==========
1. The idea of "grouping node representations using softmax for graph classification" may have been investigated in existing works, e.g., [1]. For the core idea in this paper, the authors may need to discuss its connection with existing literature, and highlight the unique perspective.

2. The technical impact of this paper could be limited.
    - From the current draft, the proposed StructAgg looks like incremental changes to the existing methods. The authors may provide more theoretical or empirical evidences to motivate the problem, highlight the uniqueness in the technique, and justify their design choices.
    - The theoretical results in Theorem 1 may need more work. Theorem 1 could aim to answer an important question in StructAgg, but the authors may raise the question first and justify why this is an important/non-trivial question.

3. The empirical evidences could be stronger.
    - From the current results, StructAgg only performs the best in one of the three datasets. The authors may consider more datasets that can highlight the value of StructAgg.
    - StructAgg may work with the existing GNNs. The authors may connect StructAgg with the existing GNNs, and demonstrate the incremental gain from StructAgg.
    - For the evaluation in Table 2, it is unclear why the comparison is limited between GCN and the proposed technique.
    - For the evaluation in Table 3, the claim on "node embedding" may not be sound. Between StructAgg and StructHist, the difference is not just "node embedding", and one cannot ignore the impact from "histogram". The authors may need to carefully reason the conclusion from the empirical numbers.

==========Questions during rebuttal period==========

Please address and clarify the weak points above.

In addition, it will be great if the authors could address the following questions.

Q1. As the authors emphasize the term "node embedding", does this paper target on transductive or inductive settings?

Q2. For the entropy minimization discussed in Section 3.3, is it going to bring more overfitting risk?

Q3. For the first condition in Definition 1, it is a bit vague. The authors may give a more accurate mathematical description.

Q4. For the first bullet in Remarks under 3.3, what does "The structural classes identified by our algorithm are local" exactly mean?

Q5. In the experiment setup, what do $l_1$ and $l_2$ refer to?

==========Reference==========

[1] Substructure Assembling Network for Graph Classification, AAAI 2018

==========Post rebuttal==========

The authors' response does not fully address my concerns. I keep the rating as it is.

---

### Official Review · AnonReviewer1 · 2020-10-29
**This work tries to identify and explore local topological roles in explainable graph learning. Writing needs improvement. Experimental results not conclusive.**

**Rating:** 3
**Confidence:** 5

**Review:**

This work tries to identify  local patterns that can be used to provide explainability to GNN models. Different from previous work using simple pooling function to generate graph representation, this work clusters nodes into a predefined number of clusters using their embeddings from L-hop neighbors. Each cluster produces a pooling representation. The representations of all clusters are concatenated to form the representation of the whole graph. However, the manuscript was not well written with some confusing notations and logical problems. It should be beneficial to readers if the manuscript can first clearly demonstrate problems in existing graph classification problems. Some analyses are not solid. The proposed method performed obviously worse than several baseline methods.

This work assumes that similar node embeddings should have similar structural roles and thus should be clustered together. Nodes with the same local structure may have the same embeddings, but not the other way around may not be true. The structural roles are defined is based on embedding similarity, which doesn’t guarantee to enhance explainability. The author(s) claimed that most GNNs were not explainable. This is not quite true. Recently, the attention mechanism was used in GNNs to enhance explainability. There exist some other works, like GNNExplainer, aiming at explainable GNNs.

The baseline models compared in this paper were proposed several years ago and not mainly focused on graph readout function. More typical methods for global pooling were missing in comparison. For instance, GlobalAttention (https://arxiv.org/abs/1511.05493) is also an explainable readout function. ASAPooling (https://arxiv.org/abs/1911.07979) is a recent work that also learns a sparse soft cluster assignment for nodes in the pooling phase. In addition, the proposed method performed worse than many baseline models in many cases. For experiments on OGB datasets, it only compares with original GCN model, but there has been a lot of state-of-art models proposed several month ago, achieving much better results than this paper. You can find the latest results on https://ogb.stanford.edu/docs/leader_graphprop/.


Some notations need improvements.
1.	Equation (2): The trainable weight matrix is missing in GCN(A, X).
2.	Definition 1 denotes two nodes as u and v, but the notation becomes i and j when representing their l-hop neighbors. X is first used as the feature matrix but later used to indicate node vector.

Some claims are not correct.
1.	In Section 3.2, “Embedding nodes with MP creates embeddings that are close for nodes that are structurally equivalent.” When many GCN layers are stacked together, final node embeddings tend to become similar.
2.	 “When computing the distance between two graphs, if those two graphs have the same distribution of structural roles, they will have embeddings that are close. ” This may be true, but the overall graph topology information is omitted. Here structural roles represent local patterns.

---

### Official Review · AnonReviewer3 · 2020-10-29
**Official Blind Review #3**

**Rating:** 7
**Confidence:** 5

**Review:**

This paper focuses on deriving explainable features for use in graph classification. To that end, they propose StructAgg that is essentially an aggregation process based on the structural roles of nodes that is then used in an end-to-end model. Experiments demonstrate the effectiveness of the proposed approach as it provides comparable performance while providing some explainability. This is an important unsolved problem, and this work provides one such approach to obtain more explainable and intuitive features/embeddings for graph classification.

The example provided in Figure 1 (a) is trivial in the sense that one can assign structural roles to nodes based on degree. For instance, red nodes all have degree 1, orange nodes have degree 3, and yellow nodes have degree 2. Please provide a better example such as the one used in the recent survey paper “On Proximity and Structural Role-based Embeddings in Networks” (see Figure 1) where this is not the case, and one must look at the actual structural properties to discover roles. Also, I suggest showing actual meaningful features in (b) as this would make the example more complete. There are a few papers related to structural roles that are missing and should be referenced appropriately. For instance, using roles for graph similarity and comparison is discussed in “Role Discovery in Networks”. There are also incorrect references to a paper from 2018 for structural roles that need to be fixed. Overall, this paper is interesting, the problem is important, and the approach is shown to be effective.

---

### Decision · Program_Chairs · 2021-01-07
**Final Decision**

**Decision:**

Reject

**Comment:**

The paper addresses an important unsolved problem, i.e. deriving explainable features for use in graph classification. It does it by providing:
i)  a simple to implement (local) node aggregation approach;
ii) some theoretical support to the proposed approach;
iii) empirical evidence that the proposed approach could be effective.

Notwithstanding the above merits, the reported work seems to still be in a preliminary phase. In fact:
i) reference to literature is missing some important recent contributions to the addressed problem (e.g.  Gated Graph Sequence Neural Networks, GNNExplainer);  if possibile, also experimental comparisons vs those approaches is desirable;
ii) experimental results do not provide a solid evidence that the proposed approach can really help to provide a clear explanation of the output, and the overall performance in classification is mostly below SOTA models; adding more datasets could help to give a more solid support to the main statement about explainability/performance;
iii) presentation needs to better highlight the original contribution w.r.t. relevant literature (which is not completely clear in the current version of the paper), to improve the explanation of proofs, to discuss (both from a theoretical and empirical perspective) some important issues, such as computational scalability with the increase of size of local structures, and robustness to noise of the proposed (local) aggregation method.

In summary, although the proposed approach seems to be of some value, more work is needed to better place the proposed approach in the context of current literature and to gain a stronger experimental support to the main claim of the paper w.r.t. explainability.